# HPV-Negative and HPV-Positive Oral Cancer Cells Stimulate the Polarization of Neutrophils towards Different Functional Phenotypes In Vitro

**DOI:** 10.3390/cancers15245814

**Published:** 2023-12-12

**Authors:** Marcela Guadalupe Martínez-Barajas, Luis Felipe Jave-Suárez, Inocencia Guadalupe Ramírez-López, Mariel García-Chagollán, José Sergio Zepeda-Nuño, Adrián Ramírez-de-Arellano, Pablo César Ortiz-Lazareno, Julio César Villegas-Pineda, Ana Laura Pereira-Suárez

**Affiliations:** 1Instituto de Investigación en Ciencias Biomédicas, Centro Universitario de Ciencias de la Salud, Universidad de Guadalajara, Guadalajara 44340, Mexico; marcebarajas.db@gmail.com (M.G.M.-B.); maye_999@hotmail.com (M.G.-C.); adrian.ramirezdearellano@hotmail.com (A.R.-d.-A.); 2División de Inmunología, Centro de Investigación Biomédica de Occidente, Instituto Mexicano del Seguro Social, Guadalajara 44340, Mexico; lfjave@gmail.com (L.F.J.-S.); pablolazareno@gmail.com (P.C.O.-L.); 3Departamento de Microbiología y Patología, Centro Universitario de Ciencias de la Salud, Universidad de Guadalajara, Guadalajara 44340, Mexico; ino_141992@hotmail.com (I.G.R.-L.); jsergio.zepeda@academicos.udg.mx (J.S.Z.-N.); julio.villegas@academicos.udg.mx (J.C.V.-P.)

**Keywords:** oral squamous cell carcinoma, human papillomavirus, tumor-associated neutrophils, polarization, functional phenotype

## Abstract

**Simple Summary:**

Tumor-associated neutrophils (TANs) participate in the initiation and development of cancer by exerting either anti-tumorigenic or pro-tumorigenic roles, depending on their functional state. Oropharyngeal cancer associated with human papillomavirus (HPV) infection is related to a better prognosis, and, interestingly, it has a lower infiltration of TANs. However, the impact of HPV’s presence on neutrophil polarization remains unknown. In this work, the effect of supernatants from oral cancer cell lines on the functional phenotype of neutrophils was evaluated. It was observed that the modulation of neutrophil polarization by tumor cells varied depending on the HPV status, with HPV− cells inducing a highly activated functional state in neutrophils, which could be associated with a pro-tumorigenic effect.

**Abstract:**

High-risk human papillomavirus (HPV) infection is one of the leading causes of oropharyngeal squamous cell carcinoma (OPSCC), while the correlation between HPV and oral squamous cell carcinoma (OSCC) remains controversial. The inflammatory infiltrate involved in these epithelial neoplasms differs based on their association with HPV. HPV− tumors show higher tumor-associated neutrophil (TAN) infiltration. It is believed that TANs can play a dual role in cancer by exerting either anti-tumorigenic or pro-tumorigenic effects. However, the impact of HPV status on neutrophil polarization remains unknown. Therefore, this study aimed to investigate the effect of OSCC cells, both HPV− and HPV16+, on the functional phenotype of neutrophils. Peripheral blood neutrophils were stimulated with supernatants from OSCC cell lines and non-tumorigenic HaCaT keratinocytes transduced with HPV16 E6/E7 oncogenes. Subsequently, cytokine production, cell viability, metabolism, expression of degranulation markers, and PD-L1 expression were evaluated. Our findings demonstrate that in contrast to UPCI:SCC154 (HPV+ OSCC) cells, the SCC-9 (HPV− OSCC) cell line induced a highly activated functional state in neutrophils, which is potentially associated with a pro-tumorigenic effect. The HaCaT 16-E7 supernatant only stimulated the activation of some neutrophil functions. Understanding the complex interplay between neutrophils and their microenvironment has the potential to identify TANs as viable therapeutic targets.

## 1. Introduction

Head and neck squamous cell carcinoma (HNSCC) is a heterogeneous group of tumors that develop from the mucosal epithelium of the sinonasal cavity, oral cavity, pharynx, and larynx [1]. According to the GLOBOCAN 2020 database, HNSCC is one of the most common types of cancer worldwide, with a high incidence and mortality rate [2,3].

Exposure to tobacco-derived carcinogens and alcohol consumption are associated with developing HNSCC [3,4]. Additionally, infection with high-risk HPV, mainly genotype 16, is implicated in the occurrence of tumors in the oropharynx [3,5]. Interestingly, besides being an etiological factor, it is also related to the disease progression. It is well known that HPV+ OPSCC cases have a better prognosis and response to treatment compared to those negative for the virus [6,7]. However, the relevance of HPV status in OSCC is less clear, and the information is controversial [8].

The immune landscape in the tumor microenvironment (TME) can either inhibit or contribute to cancer progression, depending on the type of cell populations present. In HNSCC, the TME exhibits variations according to the anatomical subsite and associated etiological agent [9,10]. Consequently, unlike HPV-unrelated tumors, cases associated with HPV infection are characterized by a high infiltration of immune cells with effector functions directed towards the destruction of the tumor, such as CD4^+^ and CD8^+^ T cells, B cells, dendritic cells, and M1 macrophages. In addition, these HPV+ tumors exhibit a low infiltration of M2 macrophages, which are related to pro-tumorigenic function [11,12], as well as TANs [12,13]. Other studies have linked the high counts of TANs to treatment resistance and an unfavorable prognosis in various types of cancer [14,15,16].

TANs are usually classified into two functional phenotypes: anti-tumorigenic N1 and pro-tumorigenic N2 [17]. Currently, there is no consensus on specific markers to distinguish them; therefore, this categorization is based on their functions. N1 has been associated with pro-inflammatory activity and characterized by high levels of TNF-α, reactive oxygen species (ROS), CXCL10, ICAM-1, and Fas. In contrast, N2 neutrophils are considered less activated [18], although some studies link them to the secretion of proteases and chemokines [17,19].

To date, it is recognized that there is a difference in the quantity of TANs based on HPV status. Unlike HPV+ OSCC, cases not associated with the virus exhibit a higher infiltration of TANs [13]. Nevertheless, it is unknown which phenotype of neutrophils predominates in each tumor type. Identifying the predominant TAN phenotype in both HPV− and HPV+ tumors and understanding the effect that tumor cells have on neutrophil polarization could clarify the differences in the immune landscape and behavior between these two types of cancer, thus potentially leading to the development of new therapeutic strategies. For these reasons, this study investigated the in vitro effect of supernatants from HPV− and HPV+ OSCC cell lines on the modulation of the functional state of neutrophils through an analysis of their cytokine secretion, cell viability, metabolism, and expression of CD66b, elastase (NE), and PD-L1.

## 2. Materials and Methods

### 2.1. Cell Cultures and Supernatant Collection

OSCC cell lines SCC-9 (HPV−) and UPCI:SCC154 (HPV16+) were obtained from the American Type Culture Collection (ATCC^®^, Manassas, VA, USA). The SCC-9 cell line was cultured in DMEM/F-12 GlutaMAX (Cat. No. 10565018, Gibco™, Waltham, MA, USA) with 10% heat-inactivated fetal bovine serum (FBS; Cat. No. 16000044, Gibco™, Waltham, MA, USA) and 1% penicillin/streptomycin (Cat. No. 15140122, Gibco™, Waltham, MA, USA). UPCI:SCC154 cells were grown in EMEM (Cat. No. 30-2003, ATCC^®^, Manassas, VA, USA) supplemented with 2 mM L-glutamine (Cat. No. 30-2214, ATCC^®^, Manassas, VA, USA), 10% FBS, and 1% antibiotic. Non-tumorigenic HaCaT keratinocytes transduced with HPV16 E6/E7 oncogenes or with the pLVX-Puro vector were provided by Dr. Luis Felipe Jave Suárez from the Western Biomedical Research Center (CIBO, IMSS, Mexico City, Mexico), and they were cultured in DMEM GlutaMAX (Cat. No. 10566016, Gibco™, Waltham, MA, USA) with 10% FBS and 1% antibiotic. All cell lines were incubated at 37 °C in a humidified air atmosphere containing 5% CO_2_.

Preparation of cell-lines-conditioned media was performed as follows: 2.2 × 10^5^ cells/well were seeded in a 6-well plate. After 24 h, the medium was replaced, and cells were further incubated for 48 h to reach 80–90% confluence. The cell-free supernatant was then collected and stored at −80 °C.

### 2.2. Neutrophil Purification, Culture, and Stimulation

The study protocol was approved by the Ethical Committee of the University of Guadalajara, and written informed consent was obtained from blood donors, following the ethical principles expressed in the Helsinki Declaration for research involving human subjects. This promotes integrity, respect, and responsibility in the field of medical science.

Peripheral blood was collected from clinically healthy subjects through venous puncture in BD Vacutainer^®^ EDTA-containing tubes (Cat. No. 367863, Becton Dickinson, East Rutherford, NJ, USA). The samples were obtained in the morning, and each independent assay in all of the methodologies was carried out with a different blood sample. The blood was separated using a two-layer density gradient formed by an upper layer of Histopaque^®^-1077 (Cat. No. 10771, Sigma Aldrich, St. Louis, MI, USA) and a lower layer of Histopaque^®^-1119 (Cat. No. 11191, Sigma Aldrich, St. Louis, MI, USA). This mixture was centrifuged for 30 min at 890× *g* and room temperature (RT). Subsequently, cells from the upper fraction (lymphocytes and monocytes) were discarded. The neutrophil-rich ring was collected, while the erythrocyte pellet remained at the bottom of the tube. The granulocytes were washed once with sterile saline solution for 10 min at 1000 rpm and RT. Neutrophils were then resuspended in RPMI-1640 (Cat. No. 11875093, Gibco™, Waltham, MA, USA) supplemented with 10% FBS and 1% antibiotic. Finally, cell viability and counting were determined with trypan blue.

Neutrophils were seeded in supplemented RPMI-1640 and incubated for 30 min at 37 °C in a humidified air atmosphere containing 5% CO_2_. Subsequently, the neutrophils were stimulated for 24 h with the supernatant previously obtained from the cell lines (diluted 1:10). Lipopolysaccharide (LPS, 100 ng/mL; Cat. No. L3129, Sigma Aldrich, St. Louis, MI, USA) was used as a positive activation control and incubation in medium alone was used as a negative control.

### 2.3. Multiplex Immunoassay

The cytokines present in the cell lines and stimulated neutrophils’ supernatants were evaluated using a multiplex immunoassay, which enables the simultaneous analysis of multiple cytokines. The Bio-Plex Pro™ Human Cytokine 17-plex Assay (Cat. No. 10023381; G-CSF, GM-CSF, IFN-γ, IL-1β, IL-2, IL-4, IL-5, IL-6, IL-7, IL-8, IL-10, IL-12, IL-13, IL-17A, MCP-1, MIP-1β, and TNF-α) and Bio-Plex Pro™ TGF-β 3-plex Assay (Cat. No. 171W4001M; TGF-β1, TGF-β2, and TGF-β3) kits (Bio-Rad, Hercules, CA, USA) were used following the manufacturer’s instructions. The cytokine concentration was determined using the MAGPIX^®^ flow system (Bio-Rad, Hercules, CA, USA) and expressed in pg/mL. A single experiment was conducted.

### 2.4. Flow Cytometry

Flow cytometry was conducted to assess cell viability and PD-L1 expression. For cell viability evaluation, we followed the manufacturer’s instructions for the FITC Annexin V and propidium iodide (PI) kit (Cat. No. V13242, Invitrogen™, Waltham, MA, USA). After 24 h of stimulation with the supernatants, 2 × 10^5^ neutrophils were collected and stained with FITC annexin V and PI. Regarding PD-L1 expression analysis, 2 × 10^5^ neutrophils were stained with anti-PDL1/PE (Mouse IgG2b; Cat. No. 329706, BioLegend^®^, San Diego, CA, USA) and incubated for 30 min at 4 °C in the dark. An isotype control antibody (Mouse IgG2b; Cat. No. 401207, BioLegend^®^, San Diego, CA, USA) was included. Subsequently, the cells were washed, and data were acquired using the Attune™ NxT Acoustic Focusing Cytometer (Invitrogen™, Waltham, MA, USA) to analyze 20 thousand events per sample with the BL1-A and BL2-A detection filters. Data analysis was performed with FlowJo™ software (version 10.5.3, FlowJo LLC, Ashland, OR, USA). Three independent assays were performed in triplicate.

### 2.5. MTT Assay

MTT assay was performed because it is one of the most useful methods for measuring cell metabolism through the evaluation of mitochondrial activity. After stimulating the neutrophils (2 × 10^5^) with the supernatant from the cell lines, 10 μL of MTT solution (Cat. No. M5655, Sigma Aldrich, St. Louis, MI, USA) at a concentration of 5 mg/mL was added, and the cells were incubated for 3 h in the dark at 37 °C in a humidified air atmosphere containing 5% CO_2_. Subsequently, the plate was centrifuged for 10 min at 2000 rpm, the supernatant was removed, and 100 μL of DMSO (Cat. No. D8418, Sigma Aldrich, St. Louis, MI, USA) was added to each well. After the formazan crystals’ complete dissolution, the solution’s absorbance was measured at 570 nm using the Multiskan™ GO reader (Thermo Scientific™, Waltham, MA, USA). Three independent assays were performed in triplicate.

### 2.6. ROS Production

To assess the production of ROS, neutrophils (2 × 10^5^) were washed once with PBS, resuspended in DCFDA/H_2_DCFDA solution (Cat. No. ab113851, Abcam, Cambridge, MA, USA) at a concentration of 20 μM, and incubated for 30 min at 37 °C in the dark. H_2_DCFDA is a fluorogenic dye that diffuses into the cell and is deacetylated by esterases, thus forming a non-fluorescent compound. This compound is subsequently oxidized by ROS to produce 2′,7′-dichlorofluorescein (DCF), a highly fluorescent molecule. Fluorescence was measured using the Synergy HTX microplate reader (BioTek, Winooski, VT, USA) with an excitation wavelength of 485/20 nm and an emission wavelength of 528/20 nm. Tert-Butyl hydroperoxide (TBHP) was included at a concentration of 55 μM as a positive control for the experiment. Three independent assays were performed in triplicate.

### 2.7. Immunofluorescence

To observe the expression pattern and quantify the CD66b and NE expression in stimulated neutrophils, immunofluorescence was conducted. First, 1.5 × 10^5^ neutrophils were adhered to electrostatically charged slides through cytocentrifugation (2 min at 2000 rpm). The granulocytes were fixed with 4% paraformaldehyde for 10 min and then washed with PBS. For NE detection, they were permeabilized with 0.2% Tween 20 for 10 min, followed by another wash with PBS. Blocking was performed with PBS/10% FBS for 1 h at RT. Then, the corresponding primary antibody was incubated for 1 h at RT: anti-CD66b (Mouse IgG; Cat. No. 392902, BioLegend^®^, San Diego, CA, USA) at a 1:50 dilution or anti-NE (Rabbit IgG; Cat. No. MA5-32548, Invitrogen™, Waltham, MA, USA) at a 1:50 dilution. The primary antibody solution was removed, the slides were washed with PBS, and then they were incubated for 1 h in the dark at RT with the secondary antibody (anti-rabbit IgG/Alexa Fluor™ 488 (Goat IgG; Cat. No A-11008, Invitrogen™, Waltham, MA, USA) or anti-mouse IgG/FITC (Goat IgG; Cat. No. ab6785, Abcam, Cambridge, MA, USA)), both at a 1:1000 dilution. A negative control group was included, wherein the primary antibody was omitted for the two secondary antibodies used. Nuclei were stained with DAPI (Cat. No. D1306, Invitrogen™, Waltham, MA, USA) at a 1:10,000 dilution for 5 min. The slides were observed under an Axio Imager A2 fluorescence microscope (Carl Zeiss AG, Jena, Germany), and 30 cells per slide were considered for fluorescence quantification using ImageJ software (version 1.53k, National Institutes of Health, Public Domain). Fluorescence was reported in arbitrary fluorescence units. Three independent assays were performed.

### 2.8. Statistical Analysis

Statistical analysis was performed using GraphPad Prism software (version 8.0, GraphPad Software Inc., San Diego, CA, USA). Data are presented as mean ± SD from the indicated number of experiments. One-way ANOVA for independent groups was conducted, followed by post hoc tests (Tukey’s, Tamhane’s T2, or Games–Howell) based on Levene’s test for homogeneity of variances. The specific post hoc test used is indicated in each figure legend. Statistical significance was considered when the *p*-value was < 0.05.

## 3. Results

Given that HPV− HNSCC tumors exhibit a higher infiltration of TANs compared to HPV+ cases, but the functional phenotype of these granulocytes is still unknown, we investigated the effect of OSCC cells on neutrophil polarization. To conduct this study, neutrophils isolated from the peripheral blood of clinically healthy subjects were stimulated with supernatants from OSCC cell lines, SCC-9 (HPV−) and UPCI:SCC154 (HPV+), with LPS as a positive control for activation, or with conditioned media from non-tumorigenic HaCaT keratinocytes transduced with HPV16 E6/E7 oncogenes or empty pLVX-Puro vector. Subsequently, cytokine production, cell viability, metabolism, and the expression of degranulation markers and PD-L1 in neutrophils were evaluated.

### 3.1. Cytokine Production by OSCC Cell Lines and Stimulated Neutrophils

Cytokines present in a TME play a crucial role in cell communication. Therefore, a multiplex immunoassay was conducted to measure cytokine production by OSCC cell lines and neutrophils. While the kits used can quantify the concentration of many analytes, only the results of the cytokines related to the N1 and N2 phenotypes are shown. In the first instance, soluble factors produced by tumor cells that could potentially be involved in neutrophil polarization were evaluated. It was observed that the SCC-9 (HPV−) cell line secreted higher concentrations of IFN-γ, IL-8, G-CSF, IL-10, TGF-β2, and TGF-β3 compared to the UPCI:SCC154 (HPV+) cells (Figure 1A). Following that, the effect of tumor cell supernatants on cytokine production by granulocytes was analyzed. Interestingly, it was found that SCC-9 (HPV−) cells promoted the secretion of high levels of all pro-inflammatory cytokines (TNF-α, IFN-γ, IL-1β, and IL-6), chemokines (IL-8, CCL2, and CCL4), and anti-inflammatory cytokines (IL-10 and TGF-β) by neutrophils, which is similar behavior to LPS. In contrast, when granulocytes were stimulated with UPCI:SCC154 (HPV+), the levels of these soluble factors were similar to the basal group (Figure 1B).

### 3.2. Stimulated Neutrophils’ Survival

It is known that HPV− HNSCC tumors exhibit a higher infiltration of TANs compared to HPV+ cases. Therefore, we evaluated whether HPV− and HPV+ OSCC cells affect neutrophil survival. Flow cytometry was performed using FITC-conjugated annexin V and PI (Figure 2A). The results show that the SCC-9 (HPV−) cell line tends to induce a higher percentage of viability, lower early apoptosis, and a more significant number of dead neutrophils compared to the response induced by UPCI:SCC154 (HPV+) cells. A similar pattern was observed in the LPS-stimulated control group in contrast to the basal group, as well as in the HaCaT 16-E7 group in comparison to HaCaT pLVX and HaCaT 16-E6 (Figure 2B).

### 3.3. Stimulated Neutrophils’ Metabolism

The polarization towards a specific functional phenotype is associated with the activation status of neutrophils, thus prompting us to assess mitochondrial activity using the MTT assay. Granulocytes stimulated with SCC-9 (HPV−) exhibited significantly higher metabolism compared to the response induced by the UPCI:SCC154 (HPV+) cell line (*p* = 0.0009). Interestingly, the effect induced by SCC-9 (HPV−) cells was similar to that of LPS compared to the basal group (*p* = 0.0073). Conversely, non-tumorigenic keratinocytes promoted an increase in the mitochondrial activity in neutrophils, although no significant differences were observed in the presence of oncogenes (Figure 3A). Additionally, to complement the evaluation of metabolism, the production of ROS by neutrophils was quantified using TBHP as a positive control for oxidation. The results obtained were consistent with those observed in the MTT assay. ROS production significantly increased after the stimulation with SCC-9 (HPV−) cells in comparison to UPCI:SCC154 (HPV+) (*p* = 0.0013), as well as with LPS compared to the basal group (*p* = 0.0265). No significant differences were found in granulocytes stimulated with the transduced HaCaT cell lines (Figure 3B).

### 3.4. Expression of CD66b in Stimulated Neutrophils

During neutrophil activation, the mobilization and release of granule content occur. Therefore, the expression of the degranulation marker CD66b was evaluated through immunofluorescence (Figure 4A). Unlike UPCI:SCC154 (HPV+) cells, the SCC-9 (HPV−) cell line significantly promoted (*p* < 0.0001) and increased CD66b expression in neutrophils, which is similar to the overexpression induced by LPS compared to the basal group (*p* < 0.0001). Remarkably, granulocytes stimulated with HaCaT 16-E7 exhibited a higher level of CD66b expression in comparison to the HaCaT pLVX group (*p* = 0.0244) (Figure 4B).

### 3.5. Intracellular Expression of NE in Stimulated Neutrophils

Degranulation can involve the release of proteases into the neutrophil cytosol or the extracellular space; therefore, the expression of NE was evaluated through immunofluorescence. In granulocytes stimulated with the SCC-9 (HPV−) cell line, a more dispersed pattern was identified compared to the granular distribution induced by UPCI:SCC154 (HPV+) cells. Similarly, the localization was more dispersed in the presence of LPS compared to the basal group, as well as in the HaCaT pLVX and HaCaT 16-E7 groups in contrast to HaCaT 16-E6 (Figure 5A). Interestingly, fluorescence quantification revealed a higher expression of NE in neutrophils stimulated with SCC-9 (HPV−) in comparison to those in the UPCI:SCC154 (HPV+) group (*p* = 0.0015). On the other hand, HaCaT pLVX cells without HPV oncogenes promoted a higher protease expression compared to the response observed when stimulated with HaCaT 16-E6 (*p* = 0.0333) (Figure 5B).

### 3.6. Expression of PD-L1 in Stimulated Neutrophils

One of the molecules mostly described for its pro-tumorigenic effect is PD-L1. Because of this, the expression of this ligand was evaluated in neutrophils stimulated with the supernatant of OSCC cells using flow cytometry (Figure 6A). It was observed that stimulation with SCC-9 (HPV−) significantly induced a higher percentage of PD-L1+ granulocytes compared to the response promoted by the UPCI:SCC154 (HPV+) cell line (*p* = 0.0002), which is similar to the effect of LPS in contrast to the group without stimulation (*p* = 0.0009) (Figure 6B).

## 4. Discussion

Within the HNSCC group, patients with HPV-related OPSCC generally show significantly better therapeutic response, progression-free survival, and overall survival [6,7] compared to HPV-unrelated OPSCC. Although the impact of HPV status on the development and clinical behavior of OSCC is controversial, some studies have reported improved survival [20] and a better prognosis when associated with HPV infection [8]. Viral etiology alone is insufficient to confer favorable behavior to HNSCC [21]. Hence, in addition to clinical differences, variations in the tumor immune landscape [11,12] could be key factors in the evolution of HPV− and HPV+ cases.

Cancer involves the formation of a complex ecosystem called the TME, in which tumor cells interact with many non-cancerous cells in an altered extracellular matrix. This microenvironment drives tumor initiation, progression, invasion, metastasis, and therapy response [22,23]. The composition of the TME in HNSCC varies among different cancer subtypes [9,10]. HPV-driven tumors, particularly in the oropharynx [21], exhibit an increased immune response characterized by higher tumor infiltration of CD4^+^ and CD8^+^ T cells, B cells, dendritic cells, and M1 macrophages, as well as a decrease in M2 macrophages and neutrophils [11,12].

Neutrophils are cells that exhibit a high functional plasticity according to their surrounding microenvironment. In particular, it is believed that TANs can polarize towards an anti-tumorigenic or pro-tumorigenic phenotype [17]. However, multiple reports have associated an increase in TANs with disease aggressiveness, therapy resistance [14], short recurrence-free survival [24], and unfavorable prognosis [15,16] in various types of cancer. Specifically, in OSCC, it is known that HPV− cases have a higher infiltration of TANs compared to HPV+ tumors [13]. Consequently, does HPV infection also affect neutrophil polarization? Could this response impact the clinical behavior of this type of cancer? Because the phenotype of TANs in both types of tumors is unknown, this study investigates the effect of HPV− and HPV+ OSCC cells on neutrophil polarization.

Cytokines present in the TME modulate the polarization of TANs. Compared to the UPCI:SCC154 cell line (HPV+), SCC-9 cells (HPV−) produced higher levels of IFN-γ (associated with the induction of an N1 phenotype) [18,25], G-CSF, IL-10, TGF-β2, TGF-β3 (related to an N2 polarization) [18], and IL-8 (a chemotactic and neutrophil-activating factor) [26,27]. It is interesting to note that although IL-8 has not been described as a factor involved in the polarization towards any TAN phenotype, previous studies have reported that HPV+ OSCC cells exhibit relatively low expression of IL-8, whereas HPV− cells increase its production [13], thus favoring the recruitment of neutrophils to the tumor site [26], increasing survival [28], and activating these granulocytes [27]. Therefore, this chemokine could be one of the key factors in neutrophil chemotaxis and polarization in OSCC.

Because the specific markers for N1 and N2 neutrophils are still unknown, we characterized the polarized neutrophils in terms of their functional state. For this purpose, primary human neutrophils from healthy donors were incubated with supernatants from OSCC cell lines. In contrast to the UPCI:SCC154 (HPV+) cell line, SCC-9 (HPV−) induced a high level of activation in neutrophils, as evidenced by the high production of pro-inflammatory cytokines, chemokines, and anti-inflammatory cytokines, increased cell viability, enhanced metabolism, overexpression of CD66b, mobilization of intracellular NE, and overexpression of PD-L1. Does this highly activated functional state correspond to an anti-tumorigenic N1 phenotype? Furthermore, does the low activation induced by HPV+ cells represent a pro-tumorigenic N2 profile?

To answer these questions, it is important to consider that the origin of the terms N1 and N2 is based on the M1/M2 nomenclature of tumor-associated macrophages (TAMs). Therefore, the classification system of TANs considers the anti-tumorigenic N1 phenotype an activated or pro-inflammatory state, and the pro-tumorigenic N2 phenotype is considered a less activated or even anti-inflammatory profile [17,18]. However, in the context of cancer, the term pro-inflammatory is not always synonymous with anti-tumorigenic, especially regarding neutrophils.

Given that neutrophils are the main mediators of the acute inflammatory response, their effector functions are aimed at destroying and containing pathogens or harmful agents, as well as facilitating the recruitment of other inflammatory cells. This activation does not imply a problem during the brief duration of an acute response; however, the situation undergoes a complete transformation when inflammation persists, wherein the pro-inflammatory role of neutrophils contributes to the development and progression of cancer [14,29].

Neutrophils stimulated with the supernatant of SCC-9 (HPV−) cells produced elevated levels of all pro-inflammatory cytokines (TNF-α, IFN-γ, IL-1β, and IL-6), chemokines (IL-8, CCL2, and CCL4), and anti-inflammatory cytokines (IL-10 and TGF-β). This high production of cytokines is related to an activated functional state in neutrophils. In this regard, TNF-α may contribute to tumor progression by acting together with TGF-β to induce endothelial-to-mesenchymal transition [30]. IL-1β is involved in regulating epithelial–mesenchymal transition, metastasis [31], and chemotherapy resistance [32]. IL-6 promotes the activation of the STAT3 pathway, thereby promoting tumor cell proliferation [33]. The overexpression of IL-8 is associated with a poor prognosis in HNSCC. In addition to promoting tumor proliferation, invasion, and migration [34], it induces a positive feedback loop to stimulate the recruitment and activation of more neutrophils. IL-10 and TGF-β play a crucial role in creating an environment that suppresses the immune system, thereby facilitating tumor growth and spread [35]. Our results, combined with the findings reported in the literature, demonstrate that several pro-inflammatory and anti-inflammatory cytokines can contribute to tumor development.

It is described that the prolonged neutrophil lifespan facilitates the secretion of pro-metastatic factors, such as oncostatin M and MMP-9, which promote the migration of tumor cells [36]. Therefore, the trend of increased cell viability and decreased apoptosis in neutrophils stimulated with the SCC-9 (HPV−) cell line could contribute to a pro-tumorigenic response.

On the other hand, the increased mitochondrial activity of neutrophils induced by SCC-9 (HPV−) cells could also be related to a pro-tumorigenic effect. While the production of ROS is one of the mechanisms of antitumor cytotoxicity, it is known that the increase in these reactive species leads to a decrease in the proliferation and dysfunction of T and NK cells [37,38], which are the main effectors of the immune response against the tumor, thus favoring cancer progression.

During the degranulation of activated neutrophils, the intracellular granular content is released either intracellularly or into the extracellular space [39]. SCC-9 (HPV−) cells stimulated the overexpression of CD66b in neutrophils, indicating granular mobilization. CD66b is a protein present on the membrane of secondary granules that is up-regulated on the cell surface during degranulation [18,39]. NE is one of the most studied proteases due to its involvement in various pathologies. The SCC-9 (HPV−) cell line promoted the release of this enzyme into the cytosol of granulocytes, visualized as a dispersed expression pattern accompanied by increased fluorescence quantification. When NE is released into the cytosol, it stimulates the ability of neutrophils to respond in a less-controlled manner and promotes the formation of neutrophil extracellular traps (NETs) [40]. The extrusion of NETs is a mechanism employed by neutrophils in microbial defense. However, in the TME, these traps function as a mesh that protects malignant cells against the attack of CD8^+^ and NK cells, thereby promoting tumor progression [41]. On the other hand, when NE is secreted into the extracellular space, it promotes metastasis by inducing angiogenesis, extracellular matrix remodeling, survival, and the escape of tumor cells through blood vessels [42].

Additionally, we observed that HPV+ OSCC cells and non-tumorigenic HaCaT keratinocytes with any of the HPV oncogenes induce different responses in neutrophils, suggesting that the activation of these granulocytes depends on the transformation state of the cell and not solely on HPV infection.

Finally, the overexpression of PD-L1 in neutrophils induced by the supernatant of SCC-9 (HPV−) cells could exert a pro-tumorigenic effect in the TME. Although the expression of PD-L1 increases in activated neutrophils after being stimulated with pro-inflammatory molecules, such as LPS [43], the hyperexpression of this ligand induces the suppression of T cells by inhibiting their proliferation and activation, which contributes to tumor progression [44]. Therefore, the use of PD-1/PD-L1 inhibitors has been included in the treatment of recurrent or metastatic HNSCC. However, not all patients respond to PD-1 blockade, highlighting the need for in-depth studies using in vitro and in vivo models to understand the interaction between tumor cells and the immune system, and, subsequently, to develop immuno-oncology alternatives to improve the clinical outcomes of HNSCC patients [45].

These findings demonstrate that the binary nomenclature of N1/N2 for categorizing TAN phenotypes may not be the most suitable approach. Some authors propose describing the state of different neutrophil populations in terms of maturity and activation [46], to which it would be relevant to add the impact that each phenotype would have on the microenvironment. Based on this and the results of the present study, we can say that HPV− OSCC cells induce a highly activated functional state in neutrophils, which could exert a pro-tumorigenic effect, unlike what happens with HPV+ OSCC cells. Thus, by modulating the chemotaxis and activation of TANs differently, HPV− and HPV+ OSCC cells promote distinct characteristics in each TME, which could lead to variations in the clinical behavior of both types of cancer and even impact the prognosis.

## 5. Conclusions

In conclusion, our current data demonstrate that HPV− and HPV+ OSCC cells induce the polarization of neutrophils towards distinct functional states, thus contributing to our understanding of the differences between both types of cancer. Nevertheless, further studies are required to explore how both HPV− and HPV+ oral and oropharyngeal cancer cells may influence different neutrophil phenotypes and potentially impact clinical outcomes. Additionally, it is important to consider other factors that influence the heterogeneity of TANs, such as sex, age, circadian rhythms, microbiota, other cellular populations present in the TME, and so forth. Integrating all of these variables will lead to a better understanding of the involvement of neutrophils in the development of HNSCC, thereby allowing TANs to be positioned as potential prognostic markers and considering not only the quantity of infiltrating neutrophils but also their functions based on their phenotype. Moreover, comprehending the predominant functional phenotype of neutrophils in HPV− and HPV+ tumors opens the door to proposing TANs as therapeutic targets. This could involve strategies designed to inhibit their recruitment, modulate their polarization, or interfere with the activation of specific effector functions associated with a pro-tumorigenic effect. As a result, personalized treatments tailored to each patient’s cancer type could be implemented to improve clinical outcomes.

## Figures and Tables

**Figure 1 cancers-15-05814-f001:**
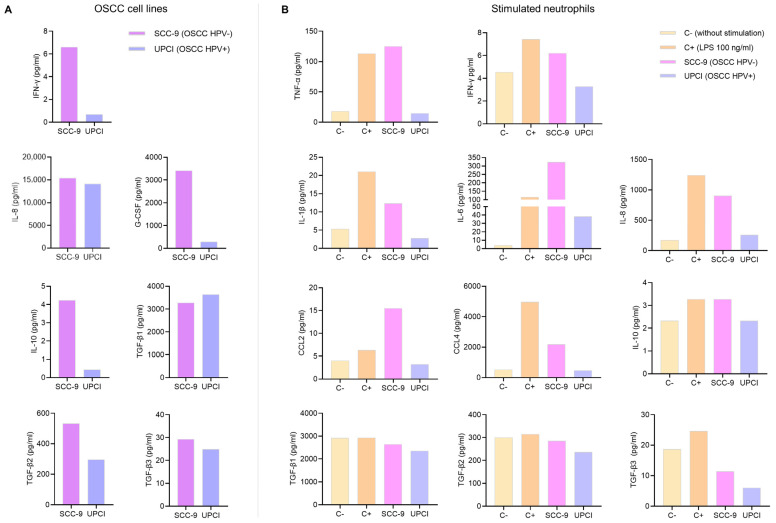
Cytokines produced by OSCC cell lines and stimulated neutrophils. Cytokine concentrations were assessed in the supernatant of OSCC cell lines (**A**) and neutrophils (**B**) using a multiplex immunoassay. The results of an independent assay are presented in pg/mL for each cytokine. C−: negative control, C+: positive control, LPS: lipopolysaccharide, pg/mL: picograms per milliliter, SCC-9: HPV− OSCC cell line, UPCI: HPV+ OSCC cell line.

**Figure 2 cancers-15-05814-f002:**
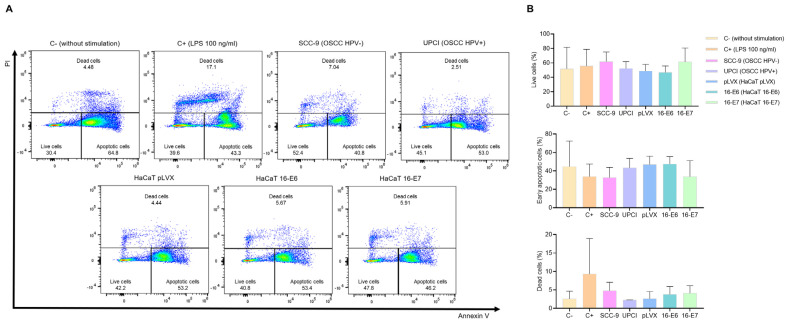
SCC-9 (HPV−) cells enhanced neutrophils’ survival. Neutrophils’ survival was assessed using flow cytometry with FITC-conjugated annexin V and PI. Representative flow cytometry panels from one of the three independent experiments (**A**) and the quantification (**B**) are shown. Data are presented as mean ± SD. No statistically significant differences were observed through one-way ANOVA. 16-E6: HaCaT keratinocytes transduced with HPV16 E6, 16-E7: HaCaT keratinocytes transduced with HPV16 E7, C−: negative control, C+: positive control, LPS: lipopolysaccharide, PI: propidium iodide, pLVX: HaCaT keratinocytes transduced with the empty pLVX-Puro vector, SCC-9: HPV− OSCC cell line, UPCI: HPV+ OSCC cell line.

**Figure 3 cancers-15-05814-f003:**
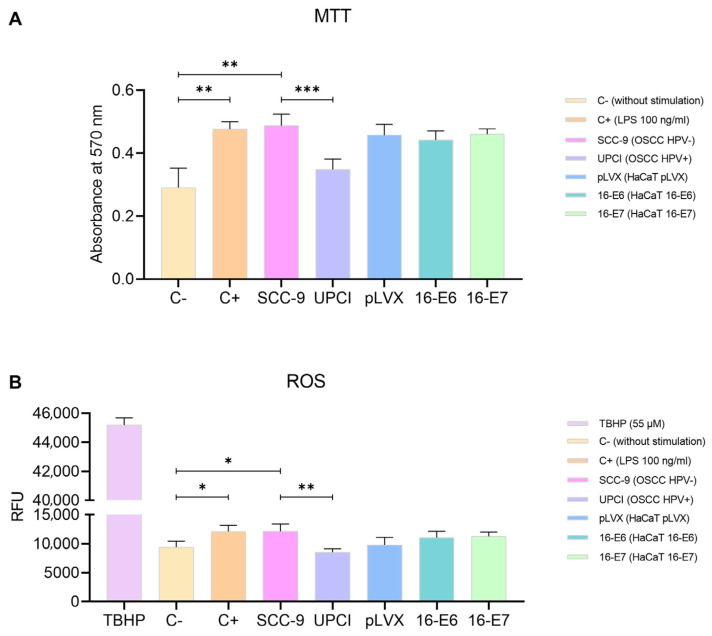
SCC-9 (HPV−) cells increased neutrophils’ metabolism. MTT assay (**A**) and evaluation of ROS production (**B**) were conducted with stimulated neutrophils. Differences were observed between C− and the other groups, except for UPCI:SCC154 (HPV+). Three independent experiments were performed in triplicate. Data are presented as mean ± SD. * *p* < 0.05, ** *p* < 0.01, *** *p* < 0.001; one-way ANOVA, post hoc test: Tamhane’s T2 (**A**) or Tukey’s (**B**). 16-E6: HaCaT keratinocytes transduced with HPV16 E6, 16-E7: HaCaT keratinocytes transduced with HPV16 E7, C−: negative control, C+: positive control, LPS: lipopolysaccharide, pLVX: HaCaT keratinocytes transduced with the empty pLVX-Puro vector, SCC-9: HPV− OSCC cell line, TBHP: Tert-Butyl hydroperoxide, UPCI: HPV+ OSCC cell line, RFU: relative fluorescence units.

**Figure 4 cancers-15-05814-f004:**
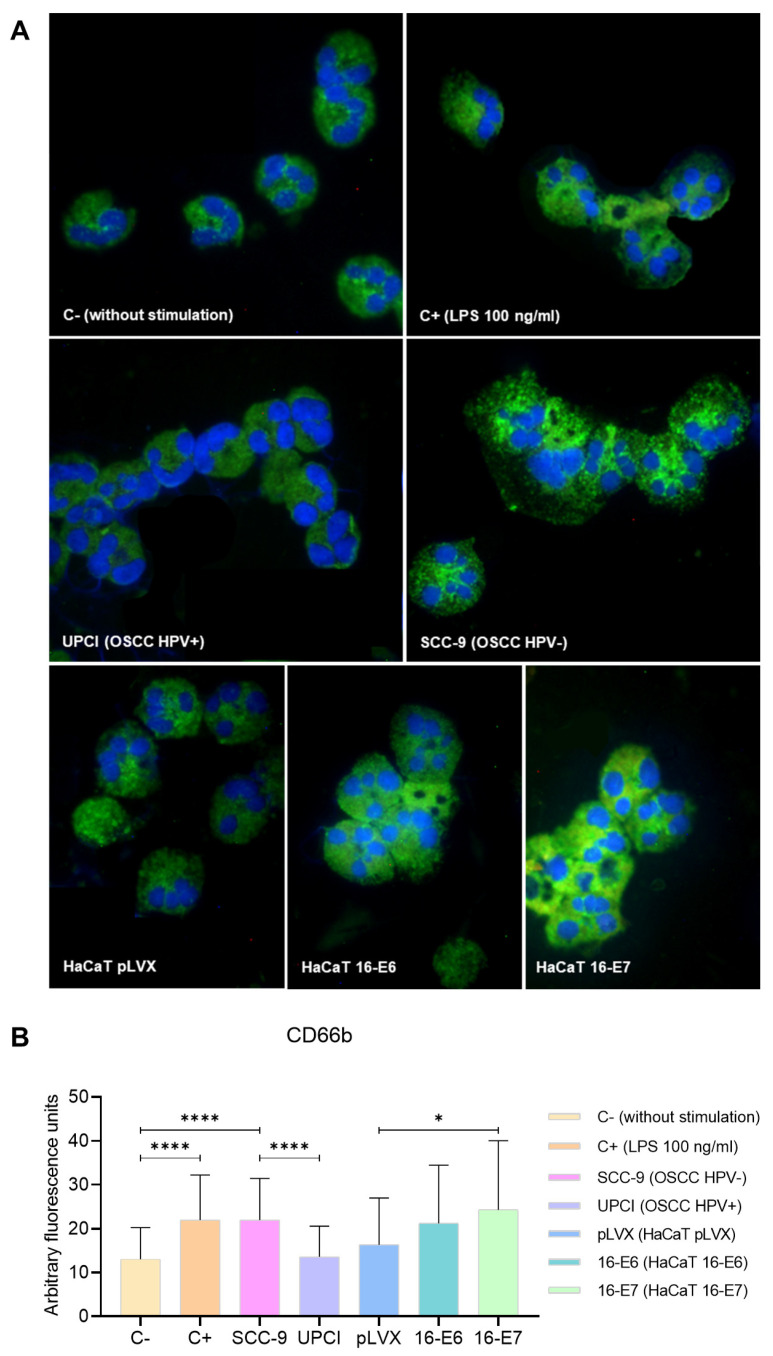
SCC-9 (HPV−) cells induced overexpression of CD66b in neutrophils. CD66b expression was evaluated through immunofluorescence using an FITC-conjugated secondary antibody (green) and nuclear staining with DAPI (blue). Merged images are shown at 40× magnification (**A**), and the quantification is presented (**B**). Differences were found between C− and the other groups, except for HaCaT pLVX and UPCI:SCC154 (HPV+). Three independent experiments were performed. Data are presented as mean ± SD. * *p* < 0.05, **** *p* < 0.0001; one-way ANOVA, post hoc test: Games–Howell. 16-E6: HaCaT keratinocytes transduced with HPV16 E6, 16-E7: HaCaT keratinocytes transduced with HPV16 E7, C−: negative control, C+: positive control, LPS: lipopolysaccharide, pLVX: HaCaT keratinocytes transduced with the empty pLVX-Puro vector, SCC-9: HPV− OSCC cell line, UPCI: HPV+ OSCC cell line.

**Figure 5 cancers-15-05814-f005:**
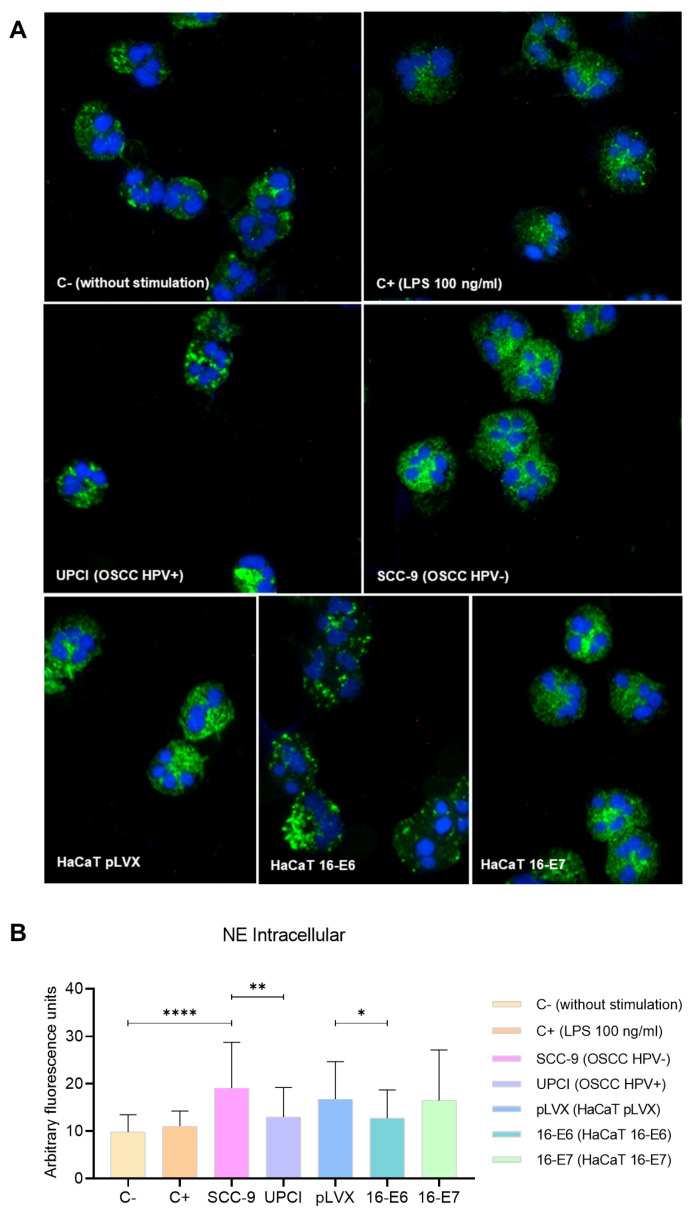
SCC-9 (HPV−) cells induced mobilization of intracellular NE in neutrophils. NE expression was evaluated through immunofluorescence using a secondary antibody conjugated with Alexa Fluor^TM^ 488 (green) and nuclear staining with DAPI (blue). Merged images are shown at 40× magnification (**A**), and the quantification is presented (**B**). Differences were found between C− and the other groups, except for C+. Three independent experiments were performed. Data are presented as mean ± SD. * *p* < 0.05, ** *p* < 0.01, **** *p* < 0.0001; one-way ANOVA, post hoc Games–Howell test. 16-E6: HaCaT keratinocytes transduced with HPV 16 E6, 16-E7: HaCaT keratinocytes transduced with HPV 16 E7, C−: negative control, C+: positive control, LPS: lipopolysaccharide, pLVX: HaCaT keratinocytes transduced with the empty pLVX-Puro vector, SCC-9: HPV− OSCC cell line, UPCI: HPV+ OSCC cell line.

**Figure 6 cancers-15-05814-f006:**
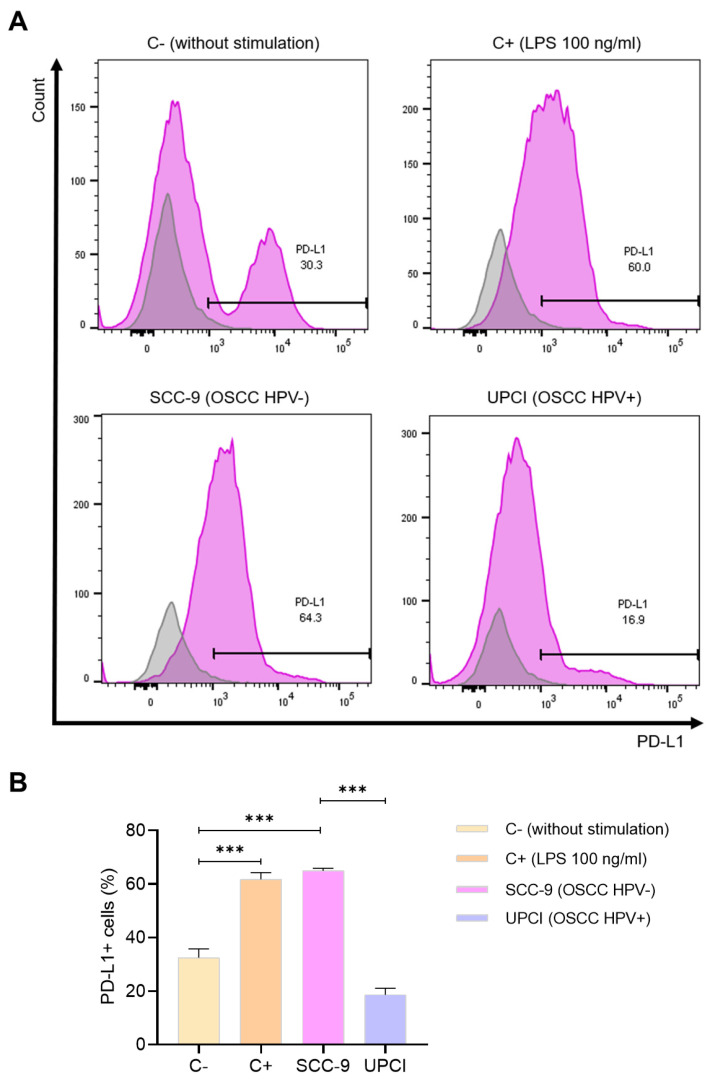
SCC-9 (HPV−) cells induced a higher percentage of PD-L1+ neutrophils. The expression of PD-L1 was evaluated using flow cytometry. Representative panels from one of the three independent experiments are shown, thus demonstrating the expression of PD-L1 (violet) alongside the corresponding isotype control (gray) (**A**), followed by quantification (**B**). Data are presented as mean ± SD. *** *p* < 0.001; one-way ANOVA, post hoc Tukey’s test. C−: negative control, C+: positive control, LPS: lipopolysaccharide, SCC-9: HPV− OSCC cell line, UPCI: HPV+ OSCC cell line.

## Data Availability

The datasets used and/or analyzed during the current study are available from the corresponding author upon reasonable request.

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
