# Peer review of "HPV-Negative and HPV-Positive Oral Cancer Cells Stimulate the Polarization of Neutrophils towards Different Functional Phenotypes In Vitro"

_cancers, 2023, doi:10.3390/cancers15245814_

Round 1

Reviewer 1 Report

Comments and Suggestions for Authors

The authors study the impact of HPV- and HPV+ supernatants on neutrophil function. The authors base the rationale of this work on the preferential accumulation of neutrophils in HPV- OSCC compared to HPV+ OSCC. They perform experiments analysing the cytokine production, survival, metabolic status and PDL-1 expression.

I have the following comments:

1. The experimental set up is not clear in the methods section. Further details are needed about the number of experiments and if each representative experiment was performed with a different donor.

2. The authors isolate granulocyte isolation from blood- what is the fraction of neutrophils in the granulocyte fraction?

3. More details are needed on the description of stimulating neutrophils with the supernatants co-culture- did the supernatants tested from the neutrophil culture include the supernatants from the cancer cell lines? If so an additional control with supernatant alone from cancer cell lines is needed.

4. Please include number of experiments performed in all Legends.

5. Stimulated neutrophils survival Annexin/PI staining: The figures do not show any significant difference so the conclusions regarding survival with HPV- supernatant are not supported.

6. MTT assay: The HaCaT lines supernatants have elicited a similar response to LPS and SCC9. Could the authors explain this observation?

7. PDL-1 expression on flowcytometry:

There are two peaks with the control cells C- on PDL-1 staining and the right peak disappears on stimulation. Are these cells dead cells with higher autofluorescence or background antibody binding? Have the authors excluded the dead cells in the flowcytometry analysis? Also does each experiment denote separate (human) donors of the tested neutrophils?

8. The authors have not explored mechanistic insights into the effects observed? What component(s) of the HPV- supernatant could potentially be exerting the effects observed?

Comments on the Quality of English Language

Some editing of the language will improve the reading of the manuscript.

Author Response

I attach the response to the requested comments

Reviewer 2 Report

Comments and Suggestions for Authors

Title:

The title accurately reflects the content of the study and is concise and informative.

Abstract:

1.      The abstract is concise and reflective

Introduction:

1.      While the introduction briefly mentions the immune landscape in the tumor microenvironment (TME) and the role of tumor-associated neutrophils (TANs), it might benefit from a more extensive explanation. Introducing this topic in more detail could help non-expert readers better understand the upcoming sections.

2.      The section discussing the N1 and N2 neutrophil profiles, while informative, is somewhat dense and might require simplification or a brief summary for clarity.

3.      The introduction could be enhanced by more explicitly stating the research objectives or hypotheses. Clearly articulating what the study aims to achieve can provide a more defined focus for the reader.

4.      The introduction lacks citations to support the mentioned facts and statistics, which is a key element in scientific writing. Including citations for statements about HPV, HNSCC, and TANs would bolster the scientific credibility of the introduction

5.      Finally, the introduction should transition more explicitly into the methodology and the specific research questions to be addressed in the study. A clear transition can help guide the reader through the paper.

Methods:

1.      While the methods are well-detailed, it might be beneficial to provide more rationale for the choice of specific techniques or reagents. Explaining why certain methods were chosen can help the reader understand the experimental design better.

2.      In scientific writing, it's essential to mention the brand or supplier of reagents and equipment to ensure transparency and reproducibility. For instance, specifying the brand of the MTT solution and the flow cytometer used would be helpful.

3.      It would be good to mention the number of replicates for each experiment, which can provide insights into the robustness of the results. For instance, how many times were the experiments conducted in triplicate, and was a single experiment conducted for cytokine analysis?

4.      While the section explains the various assays (MTT, ROS production, immunofluorescence) briefly, a bit more detail about the specific measurements taken or the significance of the assays in the context of the research would enhance the reader's understanding.

5.      There is a reference to the approval of the study by the Ethical Committee and obtaining informed consent. It would be helpful to briefly mention the ethical considerations and the purpose of this approval, especially when working with human subjects.

6.      The statistical analysis section is quite concise. It would be beneficial to mention the specific statistical tests used and how corrections were applied in more detail.

7.      The inclusion of a flowchart or diagram to illustrate the experimental workflow could enhance the reader's comprehension of the methods.

Results :

1.      The study investigated cytokine production by OSCC cell lines and neutrophils. SCC-9 (HPV-) cells secreted higher concentrations of IFN-γ, IL-8, G-CSF, IL-10, TGF-β2, and TGF-β3 compared to UPCI:SCC154 (HPV+) cells. Neutrophils stimulated with SCC-9 (HPV-) cells produced higher levels of pro-inflammatory cytokines (TNF-α, IFN-γ, IL-1β, and IL-6), chemokines (IL-8, CCL2, and CCL4), and anti-inflammatory cytokines (IL-10 and TGF-β) compared to those stimulated with UPCI:SCC154 (HPV+). Please expand on this results

Discussion:

1.      The discussion does an excellent job in breaking down the results of the study, including the differences in neutrophil responses to HPV- and HPV+ OSCC cells. The detailed description of the cytokines, chemokines, and other markers involved in neutrophil activation adds depth to the analysis. The potential pro-tumorigenic effects of these highly activated neutrophils are well-argued, considering the literature's evidence.

Conclusions:

1.      The discussion could be expanded to delve deeper into the clinical implications of these findings. How might understanding neutrophil polarization impact patient outcomes or treatment strategies?.

Language and Style:

The language used in the sections is generally clear and precise, effectively conveying scientific concepts.

Figures and Tables:

The figures and tables are thoughtfully integrated into the text, enhancing the presentation of data and correlations. They provide visual clarity and support the scientific content, effectively aiding comprehension of complex relationships within the study.

Author Response

I attach the response of the requested comments.

Reviewer 3 Report

Comments and Suggestions for Authors

Dear Editor,

Authors investigated  HPV-Negative and -Positive Oral Cancer Cells Stimulate the Polarization of Neutrophils towards Different Functional Phenotypes In Vitro. My comments are listed below;

-Abstract is ok

-Introduction is poor and it would be better to explain more about OSCC and use the below references;

*Mosaddad SA, RA Namanloo, SS Aghili, P Maskani, M Alam, K Abbasi, F Nouri, E Tahmasebi, M Yazdanian and H Tebyaniyan: Photodynamic therapy in oral cancer: a review of clinical studies. Med Oncol  40(3): 91,2023. DOI: 10.1007/s12032-023-01949-3

*Mosaddad SA, P Mahootchi, Z Rastegar, B Abbasi, M Alam, K Abbasi, S Fani-Hanifeh, S Amookhteh, S Sadeghi, RS Soufdoost, M Yazdanparast, A Heboyan, H Tebyaniyan and GVO Fernandes: Photodynamic Therapy in Oral Cancer: A Narrative Review. Photobiomodul Photomed Laser Surg,2023. DOI: 10.1089/photob.2023.0030

*Khayatan D, A Hussain and H Tebyaniyan: Exploring animal models in oral cancer research and clinical intervention: A critical review. Vet. Med. Sci,2023. DOI: https://doi.org/10.1002/vms3.1161

-Method section is ok

-Results and figures are very well

-Discussion would be better to be  explained more and use the above mentioned references.

Best wishes, 

Author Response

(The authors gave the same response as above.)

Round 2

Reviewer 3 Report

Comments and Suggestions for Authors

Dear,

My decision is acceptance.